# Comparative Cytogenetic Mapping and Telomere Analysis Provide Evolutionary Predictions for Devil Facial Tumour 2

**DOI:** 10.3390/genes11050480

**Published:** 2020-04-28

**Authors:** Emory D. Ingles, Janine E. Deakin

**Affiliations:** Institute for Applied Ecology, University of Canberra, Canberra, ACT 2617, Australia; Emory.Ingles@canberra.edu.au

**Keywords:** devil facial tumour disease, chromosome rearrangement, telomere, DNA methylation

## Abstract

The emergence of a second transmissible tumour in the Tasmanian devil population, devil facial tumour 2 (DFT2), has prompted questions on the origin and evolution of these transmissible tumours. We used a combination of cytogenetic mapping and telomere length measurements to predict the evolutionary trajectory of chromosome rearrangements in DFT2. Gene mapping by fluorescence in situ hybridization (FISH) provided insight into the chromosome rearrangements in DFT2 and identified the evolution of two distinct DFT2 lineages. A comparison of devil facial tumour 1 (DFT1) and DFT2 chromosome rearrangements indicated that both started with the fusion of a chromosome, with potentially critically short telomeres, to chromosome 1 to form dicentric chromosomes. In DFT1, the dicentric chromosome resulted in breakage–fusion–bridge cycles leading to highly rearranged chromosomes. In contrast, the silencing of a centromere on the dicentric chromosome in DFT2 stabilized the chromosome, resulting in a less rearranged karyotype than DFT1. DFT2 retains a bimodal distribution of telomere length dimorphism observed on Tasmanian devil chromosomes, a feature lost in DFT1. Using long term cell culture, we observed homogenization of telomere length over time. We predict a similar homogenization of telomere lengths occurred in DFT1, and that DFT2 is unlikely to undergo further substantial rearrangements due to maintained telomere length.

## 1. Introduction

Devil facial tumour disease (DFT1) was first detected by a photographer in 1996 in north eastern Tasmania [1]. Areas affected by DFT1 are estimated to have experienced roughly an 80% decline in Tasmanian devil population numbers [2]. The most unique trait about the disease is the transmissible nature of the tumour. As opposed to being transmitted by an infectious pathogen such as virus or bacterium, it is the cancer cells themselves which are transmitted [3]. This occurs as a side effect of Tasmanian devil social behaviour, who will often bite each other, allowing the cancer cells to transmit into open wounds [4] and thereafter growing into a new tumour. Transmitted cells are able to grow undetected by the host’s immune system, attributed in part to a downregulation of Major Histocompatability Complex (MHC) Class I gene expression on the surface of tumour cells [5]. 

Recently, a secondary and independent transmissible cancer has been found in devils, termed Devil facial tumour 2 (DFT2) [6]. This tumour was discovered as a result of histological variants in tumour samples collected in 2014 and 2015, leading to further tests to confirm the nature of these tumours. Cytogenetic comparison demonstrated significant differences between DFT1 and these tumour samples, including the presence of a Y chromosome absent from DFT1. DFT2 has a visible X and Y pair of sex chromosomes, as well as the male determining *SRY* (sex determining region Y) gene, indicating the tumour originally arose in a male [6,7]. In contrast, DFT1 has no recognisable sex chromosomes, no Y genes, but on average 2 copies of X chromosome genes, indicating that it arose in a female devil [8,9]. Further genetic tests using microsatellites, detection of somatic structural variants previously identified in DFT1, and differences in MHC class 1 genetic structure all indicated a second and independently derived transmissible tumour had arisen in the Tasmanian devil population [6].

There are marked differences in chromosome rearrangements between DFT1 and DFT2, with DFT2 having undergone relatively simple rearrangements compared to DFT1 [7]. Both have a 2n = 13 karyotype, compared to the 2n = 14 karyotype for Tasmanian devils, yet are otherwise dissimilar. The major rearrangement proposed to have led to the highly rearranged DFT1 karyotype was the end-to-end fusion of one homologue of chromosome 1 to the maternal copy of the X chromosome. This fusion was most likely followed by a series of break–fusion–bridge cycles, which led to the distinctive DFT1 karyotype, where a homologue of chromosomes 1, 4, 5, and both copies of the X chromosome are unrecognisable due to extensive rearrangement [9,10,11]. In comparison, in DFT2 the most notable rearrangement is the translocation of a chromosome 6 homologue into a chromosome 1 homologue [7]. Minor karyotype differences within DFT1 have arisen over time, leading to at least four distinct karyotypic lineages [12]. No such lineages have yet been described for DFT2. Overall, the DFT2 karyotype is more visibly similar to the Tasmanian devil karyotype compared to DFT1. 

Interestingly, the key major rearrangement for both tumours appears to have arisen from the fusion of a chromosome, with potentially eroded telomeres, to another chromosome. The unique feature of devil and other dasyurid chromosomes is the presence of a telomere length dimorphism, wherein their maternally derived chromosomes appear to have shorter telomeres, and paternal chromosomes have longer telomeres [13]. The maternal X chromosome, the X homologue with short telomeres, fused to chromosome 1 in DFT2 [10,11] and the homologue of chromosome 6 with short telomeres translocated to chromosome 1 in DFT2 [7]. Whilst the telomere length dimorphism is not observed in DFT1 [13], it is in DFT2 [7], leading us to hypothesise that a similar fate awaits the telomeres of DFT2, wherein the telomere lengths will become more homogenized.

The discovery of DFT2 provides a unique opportunity missed with other transmissible cancers. The original devil facial tumour (DFT1) only began to be investigated in earnest a decade after first being reported [1]. Similarly, other wild occurring transmissible cancers are likely to have existed for many years before discovery. Canine transmissible venereal tumour has existed for up to 11,000 years [14,15,16]. Soft shell clams are affected by a transmissible leukemia, termed disseminated Neoplasia [17], which was identified as a disease 40 years ago in the late 1970s, but could potentially be much older [18,19]. Little is known about how any of these cancers have changed or adapted since their emergence until the current day, due to a lack of information from when they were first discovered. Comparisons between the older (DFT1) and younger (DFT2) transmissible tumours provide an opportunity to investigate the evolution of transmissible tumours and to make predictions about the evolutionary fate of the younger tumour. We used molecular cytogenetic mapping to provide more detailed information on the extent of rearrangement of DFT2 chromosomes at a cytogenetic level. We compared gene arrangement between DFT2 tumour cell lines established from different individuals to observe how DFT2 is evolving. We performed long-term cell culture experiments, up to 200 population doublings (pd), to investigate changes in the telomere landscape over time, and predict whether DFT2 might also undergo a homogenization of telomere length. We also investigated broad patterns of DNA methylation using immunofluorescence between DFT2 strains, and within the long-term cell culture to determine whether there are any changes in global DNA methylation patterns. 

Our aim was to understand and predict the past and future evolution of DFT2. We propose that a silencing of the centromere of the rearranged chromosome 6 in DFT2 circumvented the necessity for further rearrangement through breakage–fusion–bridge cyles that led to the highly rearranged DFT1 chromosomes. Rearrangements in both DFT1 and DFT2 have involved chromosomes with short telomeres, implicating these as an important mechanism in the formation of these tumours. DFT2 is evolving, and at least two karyotypic lineages can be detected by gene mapping. We predict that, over time, the telomere length dimorphism will homogenize in DFT2, as occurred in DFT1, and we anticipate that maintenance of telomere lengths will reduce the chance of further substantial rearrangements in DFT2.

## 2. Materials and Methods 

### 2.1. Preparation of DFT2 Metaphase Chromosomes

DFT2 cell lines TD_500, TD_549, and TD_554 provided by Professor Greg Woods (University of Tasmania) were grown in AmnioMAX-C100 (Thermo Fisher Australia Pty Ltd, Scoresby, Victoria, Australia) at 35 °C with 5% CO_2_. Metaphase chromosomes were harvested following synchronization with colcemid (10 μg/mL) for 1 h, incubation for 15 min at 37 °C in 33% (*v*/*v*) fetal calf serum. Metaphases were fixed with chilled methanol:acetic acid (3:1) according to standard technique. Cell suspensions were dropped onto slides, air-dried, dehydrated through a 70% (*v*/*v*), 90% (*v*/*v*), 100% (*v*/*v*) ethanol series (3 min in each) and aged overnight at 37 °C prior to hybridisation. Prepared metaphase chromosomes for the cell line RV were provided on slides by Kate Swift (Department of Primary Industries, Parks, Water and Environment; Tasmania). 

### 2.2. Fluorescent in Situ Hybridization (FISH)

Bacterial artificial chromosomes (BAC) clones distributed across all six autosomes and the X chromosome were selected as probes for FISH from those previously mapped to devil chromosomes (Appendix A) [9] or those isolated from the BAC library to fill in mapping gaps on chromosomes 2 and 5 identified in the previously published devil cytogenetic map [9].

To screen the VMRC-49 devil BAC library, overgo probes were designed for genes predicted to fall within the mapping gaps identified on chromosomes 2 and 5. Genomic sequences for the genes of interest were obtained from the Ensembl Devil_ref7.0 assembly for input into the Overgo Maker program developed by The Genome Institute at Washington University. The specificity of each 40 bp overgo probe was checked by BLASTN searches of the devil genome assembly to ensure that only probes with a single match across the entire 40 bp were used for BAC library screening. Overgos used for screening are listed in Appendix A. VMRC-49 BAC library filters were screened with a pool of radioactively labeled overgo probes as previously described [20,21]. End sequencing was performed on BAC clones by Macrogen (Korea) using primers pCC1™/pEpiFOS™ forward sequencing primer (5′ GGATGTGCTGCAAGGCGATTAAGTTGG 3′) and pCC1™/pEpiFOS™ reverse sequencing Primer (5′ CTCGTATGTTGTGTGGAATTGTGAGC 3′). The BAC end sequences (Appendix A) were used as queries in BLASTN searches of the devil genome assembly to confirm the BAC contained the gene of interest. 

DNA from each BAC clone was extracted using the WIZARD Plus SV Minipreps DNA Purification System (Promega, Alexandria, NSW, Australia) with a slightly modified protocol where 15 mL of pelleted BAC culture was resuspended, lysed and neutralized in double the volume recommended by the manufacturer of each corresponding solution, and resulting supernatant was added to the WIZARD spin column in two steps rather than one. Subsequent washing of the column and DNA elution was carried out following the manufacturer’s protocol. Alternatively, BAC DNA was extracted using the PhasePrep BAC DNA Kit (Sigma-Aldrich Pty Ltd, Castle Hill, NSW, Australia) following the manufacturer’s protocol. BAC DNA was directly labeled by nick translation with either Green or Orange dUTP (Abbott Molecular Inc., Des Plaines, IL, USA) and hybridized to metaphase chromosomes following the previously described protocol [22]. Slides were washed to remove unbound probe using the previously described protocol [9] and were mounted in Vectashield with DAPI (4’,6-diamidino-2-phenylindole) (Vector Laboratories Inc., Burlingame, CA, USA). Fluorescent signals and DAPI stained chromosomes were visualized on a Zeiss Axiocam Scope A1 epifluorescence microscope with an AxioCam Mrm Rev 3 CCD (Carl Zeiss Ltd, Cambridge, UK) and using ISIS Fluorescence Imaging System software version 5.4.11 (Metasystems, Newton, MA, USA). At least 10 metaphase spreads for each hydridization were captured for analysis. Flpter (fractional length p arm terminus) values [23] were determined by measuring chromosomes and position of signals using Adobe Photoshop CC version 14.0 (Adobe Systems, Inc., San Jose, CA, USA). 

### 2.3. C-Banding

DFT2 chromosomes for three DFT2 cell lines (TD_500, TD_549, and TD_554) were C-banded using the C-bands by Barium hydroxide using Giemsa (CBG) method [24,25]. Briefly, slides of fixed metaphase chromosomes that had been aged at room temperature for 2 days were incubated in 0.2 N HCl for 40 min, rinsed in deionized water and denatured in 5% Ba(OH)_2_ for 10 min at 50 °C. Slides were then rinsed in 0.2 N HCl and deionized water before renaturing in 2xSSC (Saline Sodium Citrate) for 60 min at 60 °C. Chromosomes were stained with 4% Giemsa in 0.1 M phosphate buffer for 30 min at room temperature. Stained slides were rinsed in deionized water, air-dried, and mounted with DPX mounting medium (Sigma-Aldrich) and viewed under bright-field on a Zeiss Axiocam Scope A1 epifluorescence microscope. 

### 2.4. Long Term Cell Culture of DFT2 Cells

DFT2 TD_554 cells were used for long term cell culture. These cells were cultured in Gibco AmnioMAX C-100 medium in T25 cm^2^ flasks, at 35 °C with 5% CO2. After reaching confluence, cell cultures were washed with 1 mL of Gibco phosphate buffered saline pH 7.4 (PBS), treated with 1 mL of Gibco 0.25% trypsin-EDTA and gently agitated, 9 mL of new AmnioMAX medium was added, and cells were divided into two new flasks. DFT2 cells were cultured up to a population doubling (pd) of 200. Cells were harvested and metaphase chromosomes prepared (as above) at pd of 5 and 200 after a three and a half hour incubation with 0.1 µg/mL of colcemid (Gibco Karyomax in Hanks’ Balanced Salt Solution; Thermo Fisher Australia Pty Ltd, Scoresby, Victoria, Australia).

### 2.5. DNA Methylation Immunofluorescence

Slides were denatured in 70 % (*v*/*v*) formamide in PBS for 1 min 40 s at 70 °C, then washed in ice cold (−20 °C) 70 % (*v*/*v*) ethanol for 5 min, followed by a wash of 90% (*v*/*v*) ethanol then 100% (*v*/*v*) ethanol, both at room temperature for 3 min each. Slides were briefly air dried before being rehydrated in PBS-T (PBS with 0.03 % (*v*/*v*) Polysorbate 20) for 3 min. Slides were washed for 20 min in 1% (*w*/*v*) Bovine Serum Albumin in PBST. Slides were incubated for 1 h at 37 °C with the anti-5-methylcytosine (Clone 10G4) (Zymo Research, Irvine, CA, USA), diluted 1:200 in PBST, before being washed twice for 5 min in PBST at room temperature. Slides were incubated for 1 h at 37 °C with anti-mouse Cy3 antibody, diluted 1:500 in PBST, then washed twice for 5 min in PBST at room temperature. The slides are then fixed in 4% (*w*/*v*) paraformaldehyde in PBS for 15 min at room temperature, before being washed 3 times in PBST for 3 min at room temperature. Slides are air dried, then mounted with DAPI in Vectashield (Vector Laboratories Inc., Burlingame, CA, USA). Fluorescent staining was visualized on a Zeiss Axiocam Scope A1 epifluorescence microscope. Images of 20 metaphase spreads from each sample where captured using an AxioCam Mrm Rev 3 CCD (Carl Zeiss Ltd, Cambridge, UK). Line scans of individual chromosomes were produced using the ImageJ (1.52q) [26] RGB profile plot of the intensity of green (5-methylcytosine) and blue (DAPI) pixels on a line drawn along the length of the chromosome.

### 2.6. Telomere PNA and Telomere Length Analysis

After undergoing 5-MeC staining, slides were rinsed in 2xSSC for 2 min at room temperature and then underwent PNA FISH. Afterwards, these were observed under the fluorescence microscope in the same microscopy session, to account for any changes in light intensity without using a standard. 

Telomere PNA FISH was performed as previously described [27]. Images were captured using an AxioCam Mrm Rev.3 charged-coupled device camera (Carl Zeiss Ltd, Cambridge, UK) and ISIS Fluorescence Imaging System software version 5.4.11 (Metasystems, Newton, MA, USA). For analysis of telomere length, all images were captured with an exposure time of 3 s. Unedited images were exported as tagged image format (TIF) files and analysed using TFL-Telo telomere measurement and analysis software [28] to measure telomere length for individual chromosomes. Measurements were written in Excel and then transferred to R software for analysis. For statistical values, we used Welch’s ANOVA and the Games-Howell post-hoc tests, due to broken assumptions of normality and homogeneity of variance.

## 3. Results

We investigated DFT2 evolution in two ways, firstly by comparing gene order on DFT2 chromosomes from four different tumours and secondly by examining telomere length after 200 population doublings in vitro. 

### 3.1. Chromosome Rearrangements in DFT2

We selected 57 BAC clones, spread across the autosomes and the X chromosome (Appendix A), to generate cytogenetic maps of DFT2 chromosomes. We compared the locations of these 57 BACs on chromosomes prepared from four tumours taken from different individuals to identify any signs of tumour evolution. 

We confirmed the previously noted addition of chromosome 6 to one copy of chromosome 1 [6,7]. However, we detected differences in gene arrangement between samples obtained from different individuals for the translocated chromosome 6 material. Tumours RV and TD_549 are similar to each other with chromosome 6 material only on one side of the centromere, whereas tumours TD_500 and TD_554 have chromosome 6 genes on either side of the centromere (Figure 1a–h). We mapped genes *PDE5A, SLC1A2, DMRT1,* and ENS8760 from this chromosome 1 and 6 fusion region to chromosomes from TD_554 grown in long-term culture to see if the region was prone to continued rearrangement. Gene order was maintained in this region after 200 population doublings (Figure 1i,j). The comparative gene mapping results determined that the translocated chromosome 6 was originally inserted just above the centromere on the short arm of chromosome 1 and likely to have undergone at least one inversion, leading to the current arrangement in cell lines RV and TD_549 and a further two inversions to result in the arrangement observed in TD_500 and TD_554 (Figure 1k). C-banding was used to determine whether there was a heterochromatic band outside of the centromere constriction on the derived DFT2 chromosome 1, corresponding to the centromere from the translocated chromosome 6. A heterochromatic band is visible on the long arm of TD_549 (Figure 1l) but on the short arm of TD_500 and TD_554 (Figure 1m,n).

Other rearrangements on the der(1)t(1;6) chromosome include the addition of segments of chromosome 3 as well as the duplication of the chromosome 1 short arm gene *RAMP3* to the distal end of the long arm of this chromosome (Figure 2), identifying a copy number gain not identified from tumour sequencing (Appendix A) [7]. A similar duplication of the long arm gene *OSBP2* to the telomeric end of the short arm but this duplication was also not detected by sequence analysis (Figure 2). Apart from these rearrangements, gene order resembles that of devil chromosome 1 [10] and that of the other DFT2 chromosome 1. 

The long arm of one homologue of chromosome 2 in DFT2 tumours is longer than the other. In all four tumours examined, a copy of *TCERG1L* was missing from this region. In RV, TD_500 and TD_554, this region contained genes duplicated from the short arm of chromosome 2 (*ADRA1A, R3HCC1, ADAM7*) (Figure 3). These copy number loss and gains had previously been detected by analysis of DFT2 sequence (Appendix A) [7]. An additional difference was detected in TD_549, where one copy of *FSHR* has translocated to chromosome 1 (Figure 3) but both copies of this gene are located on the short arm of chromosome 2 in the other three tumour cell lines. 

A consistent size difference between chromosome 3 homologues was also detected, as well as the translocation of the short arm gene *DCUN1D1* to the long arm of chromosome 3 homologue (Figure 4a,b). In addition, the chromosome 1 gene *NCLN* is present not only on both copies of chromosome 1 but on both homologues of chromosome 3 (Figure 4c), a copy number gain not detected by sequence analysis. The only other difference between devil and DFT2 chromosomes was detected for one homologue of chromosome 5, which was missing copies of long arm genes *FGL2* (Figure 4d) and *ADARB2* (Figure 4e). These copy number losses were also detected by sequence analysis (Appendix A) [7].

We analysed anti-5-methylcytosine antibody staining to determine whether rearrangements impacted on global DNA methylation (5′methylcytosine). Broad patterns of DNA methylation were similar to those observed in past Tasmanian devils and DFT1 metaphases observed [11,26], where strong methylation signals at the ends of chromosomes and light segmented staining throughout the chromosome body were observed. Variability in 5′methylcytosine staining between metaphase spreads from the same sample (same slide) made it difficult to determine if there were differences in methylation patterns between the three DFT2 tumour cell lines TD_554, TD_549, or TD_500 (Appendix A) or between pd 5 and 200 for TD_554. 

### 3.2. Telomere Length in DFT2 over Time

We analysed telomere length change in the context of telomere length dimorphism, which defined the devil telomeric landscape. For chromosomes prepared at pd 5, there is a large variance between the ‘short’ and ‘long’ subset of telomeres (Figure 5). How these change over time is dependent to which subset they belong, with initial mean telomere fluorescence unit (TFU) being a strong predictor for change in TFU (Appendix A). Long telomeres undergo a drastic reduction in telomere length, an average decrease of −3787 TFUs. Short telomeres instead see a more modest increase in telomere length averaging 602 TFUs. This effect is diminished on both the X and Y chromosomes, which have both the smallest increase and decrease in TFU for their length groups respectively (22 for X and −1664 for Y), and also the least significant changes for their groups (Table 1). Welch’s ANOVA found significant differences between our groups (*p* < 2 × 10^−16^), summarized by the Games-Howell post-hoc test (Table 1). Overall telomere length has begun to homogenize by pd 200, with telomere length dimorphism being far less pronounced (Figure 5). Corroborating the increase in short telomere length, we also notice a decrease in the total number of signals below detection limits over time. Short telomeres begin with almost a third (32.2%) of their telomere signals below detection limits at pd 5, but this number decreases to 7.9% by pd 200. Conversely, the number of signals below detection limits in the long telomere group is relatively low at both time points, but with a slight decrease over time as well (1.1% to 0.4%, Table 2).

## 4. Discussion

We have combined information from cytogenetic mapping and telomere length analysis to provide insight into the evolution of DFT2 tumours and predict similarities and differences in the evolutionary trajectories of DFT1 and DFT2. 

### 4.1. Comparison of DFT1 and DFT2 Origin and Evolution

A comparison of gene order on the derived chromosome 1 in DFT2 suggests that it originated from the fusion of chromosome 6 into a position just above the centromere on chromosome 1. No interstitial telomeric signals were detected suggesting that the chromosome 6 telomeres may have already eroded. This interpretation provides a mechanism for the triggering of its rearrangement and potentially the carcinogenesis of DFT2, and also fits with our understanding that the rearranged chromosome 6 belonged to the ‘short’ telomere set of chromosomes. Although the translocated chromosome 6 has undergone at least two inversions during the evolution of DFT2 (Figure 1), the chromosome has not experienced the extensive rearrangement observed on the chromosome derived from an end-to-end fusion of chromosomes 1 and X in DFT1. We suggest that the difference between DFT1 and DFT2 in the level of rearrangement of these derived chromosomes is due to the fate of the second centromere. In DFT1, the two centromeres on the dicentric chromosome resulted in a series of breakage–fusion–bridge cycles that persisted until the second centromere was lost and resulting in an extensively derived chromosome (Figure 6). In contrast, we posit that the centromere derived from chromosome 6 in DFT2 was inactivated relatively early in the history of this dicentric chromosome, leading to stabilisation of the chromosome. 

Although the karyotypes are for the most part, relatively stable, DFT1 and DFT2 both show signs of karyotypic evolution. We detected evolution of DFT2 chromosomes 1 and 2 occurring. There are clearly at least two distinct karyotypic lineages of DFT2, where the most notable difference is position of the translocated chromosome 6 in regards to the centromere. One lineage is represented by tumours RV and TD_549, where the chromosome 6 material is on one side of the centromere. The other lineage is represented by TD_500 and TD_554, due to at least one inversion occurring since these two lineages diverged. The absence of further rearrangement of this region after long-term culture of TD_554 suggests the translocated region is now stable. 

Evolution of karyotypically distinct lineages of DFT1 has been implicated in differences in transmission and epidemiology of the disease [29]. It will be important to monitor DFT2 lineages for any differences in tumour transmission and impacts on epidemiology. At this stage, DFT2 has a limited geographic distribution and a seemingly low infection rate compared to DFT1 [30]. The karyotypic difference between the two DFT2 lineages is difficult to detect without the use of molecular cytogenetics to distinguish the position of the chromosome 6 genes, meaning that monitoring of these two lineages as they spread through the population will require continued use of FISH probes on metaphase chromosomes. Alternatively, a PCR-based diagnostic test may be able to be developed if sequence analysis is able to detect the diagnostic rearrangement breakpoints. 

### 4.2. Telomere Length in DFT2 Cell Culture

In DFT2, we provide evidence that telomere length has been homogenizing in cell culture over a time span of 195 population doublings. This is especially apparent on the longer subset of telomeres, which have a marked reduction in TFU. The shorter subset of telomeres are not only maintained, but show a slight increase in average TFU, as well as a drastic reduction in the relative number of signals below detection limits (Table 2). Early experiments using telomere PNA FISH in human fetal liver, adult bone marrow, and myeloid leukemia cells observed a similar phenomenon, wherein telomere lengths had a skewed distribution with bias towards short, but not critically short, telomeres, suggesting either a selection bias towards ‘healthy cells’ without critically short telomeres, or that short telomeres were being maintained [31]. Our results are consistent with, and predicted by, current knowledge and theory on telomeres. Telomeres are eroded due to cell divisions [32,33] and oxidative damage [34,35]. Due to the larger area of longer telomeres, these are more likely to be targeted by free radicals, causing longer telomeres to have faster attrition rates [36,37]. Conversely, the telomere lengthening enzyme telomerase will preferentially target the shortest telomeres [38,39,40]. Together, these would suggest an effect of telomere length homogenizing over time in cells with sufficient telomerase activity, wherein the longer telomeres are shortened, and the shorter telomeres are either maintained or slightly lengthened. On top of our own results, this effect has been observed in other studies [41,42].

The telomere length homogenization we observed provides a reasonable explanation for what happened in DFT1. It is probable that DFT1 originally exhibited the telomere length dimorphism, which would have been present in the original host animal, but this dimorphism disappeared over time due to homogenizing effects by the time researchers investigated its telomere distribution [13]. That we observe telomere length dimorphism in DFT2, but not DFT1, may be indicative of having caught DFT2 tumours early on in their long-term potential spread and development. Given enough time, we predict that the wild DFT2 tumours will undergo the same telomere length homogenization process. If this is true, it would provide evidence of homogenization of telomere length occurring in vivo. Homogenization effects have generally been observed in vitro, where homogenization occurs over time during cell culture conditions [41,42,43]. 

Telomere measurements taken in vivo tend to have more noticeable heterogeneity [44,45]. This is probably a result of two things. The first is the effects on telomerase expression and telomere shortening in cell culture. Most highly proliferating cells used in cell culture will express higher telomerase levels [46], and this is especially true for immortalised or cancer cells [47]. This is exacerbated by the increased proliferation of cells in cell culture, itself associated with an increase in telomerase expression [48]. Further, the in vitro cell culture environment provides a much more ‘stressful’ and pro-oxidant environment than in vivo conditions, which is likely to affect the shortening rate of telomeres [49]. The second is that telomere length heterogeneity is partially established both by heritability of telomere lengths from contributing parents [50,51,52,53,54], but also by processes either in the germline or during development that may further alter telomere length and heterogeneity [55,56,57]. DFT1 and DFT2 are likely to be unusual cases due to their cancerous nature, possessing a much more rapid growth of cells than usual and bypassing normal reproductive processes. This could explain why homogenization of telomere lengths was observed in DFT1 when it is rarely observed in live animals or individuals.

Interestingly we also noticed a diminished change in TFU in both sex chromosomes, particularly the Y. For both the ‘short’ and ‘long’ subgroups, they have either the smallest increase or decrease in TFU over time. This observation could simply reflect the heterogeneity of telomere dynamics of individual chromosomes [58], but it is suspicious that the Y chromosome stands so far apart from other ‘long’ telomere chromosomes in its shortening rate. What differentiates the Y chromosome from other chromosomes is a lack of visible methylation using immunofluorescence staining. All other chromosomes in DFT2 (Appendix A), DFT1 and Tasmanian devils have noticeable DNA methylation at the proximal ends of chromosomes [11,27], yet little to no signal is observed on the Y (Appendix A). Could the methylation status of subtelomeric DNA be playing a role in telomere attrition rates for the DFT2 Y chromosome? Subtelomeric methylation is known to regulate the homeostasis of telomere length [59]. In mice, a lack of subtelomeric methylation led to an increase in telomere length, as a result of increased recombination [60]. Yet in human cells increased DNA methylation at subtelomeric loci has been associated with increased leukocyte telomere length [61], or has been associated with increased telomere length as a result of artificially increased subtelomeric methylation [62]. It is difficult to draw parallels with these findings. Does the Y chromosome have increased levels of recombination which offset the shortening rate? Ultimately, we need more targeted information about the subtelomeric methylation of Tasmanian devil and DFT2 chromosomes to ascertain whether this is playing a part in the homeostasis of the Y chromosome’s telomeres, and whether this is specific to DFT2 or Tasmanian devils in general.

Cytogenetic mapping has allowed us to predict the processes of chromosomal rearrangements that led to DFT1 and DFT2. Unlike the more rearranged DFT1, DFT2 has avoided significant rearrangement, such as through a breakage–fusion–bridge cycle. We propose that this is due to the silencing of the rearranged chromosome 6 centromere. We further detect two distinct lineages of DFT2 using gene mapping. This has parallels to DFT1, which has at least four distinct lineages based on minor karyotype differences [12]. There is evidence that these lineages can evolve independently, with one lineage having undergone human induced selection on the Forestier Peninsula in Tasmania, leading to a predominantly tetraploid lineage that is able to escape detection for longer [63]. Monitoring of the spread of DFT2 and its lineages will be necessary to track the evolution of DFT2. Developing a tool, such as a PCR test, to distinguish these lineages without using FISH would make it easier to track the spread of the lineages. Our in vitro cell culture experiments predict that telomere lengths in DFT2 are not only likely to homogenize over time, but will also become stable and be maintained. Without telomere length shortening to induce chromosomal instability, there is a significantly reduced chance of further major rearrangements occurring in DFT2. DFT1, which has also maintained its telomeres, has had no major rearrangements in the time it has been observed, and we may expect DFT2 to be similar in that respect.

## Figures and Tables

**Figure 1 genes-11-00480-f001:**
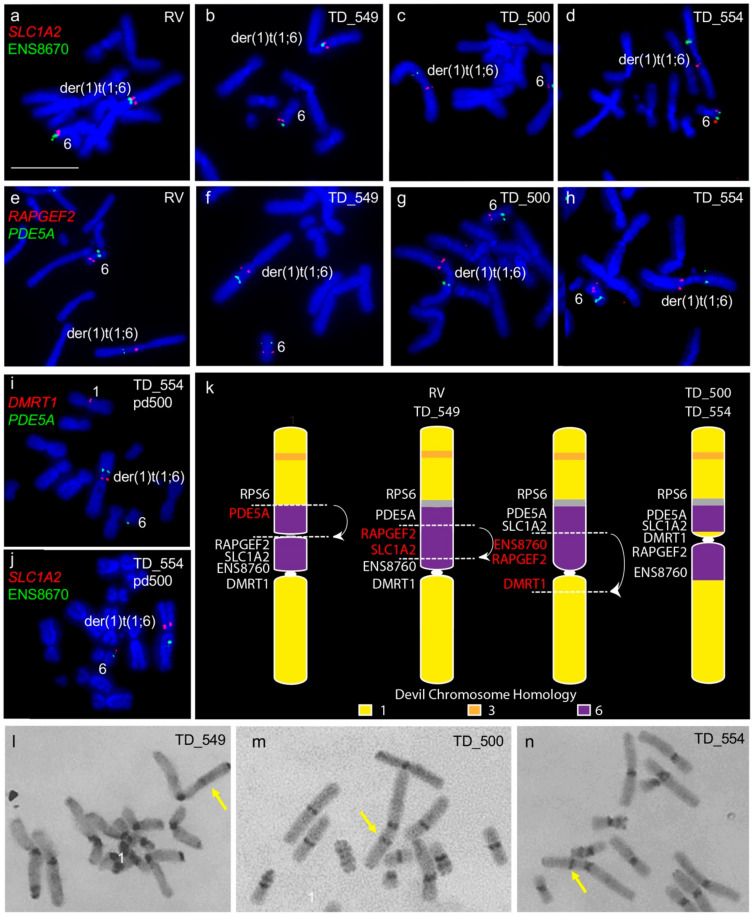
Rearrangement of translocated chromosome 6 region between all four tumour cell lines. (**a**–**d**) Colocalisation of *SLC1A2* (red) and ENS8670 (green) and (**e**–**h**) *RAPGEF2* (red) and *PDE5A* (green). Genes (**i**) *DMRT1* (red) and *PDE5A* (green) and (**j**) *SLC1A2* and ENS8670 (green) mapped onto TD_554 chromosomes at pd 200 demonstrating stability of the rearranged regions. (**k**) Tracing the evolution of der(1)t(1;6) chromosome in DFT2. At least three inversions have taken place since the fusion of translocation of chromosome 6 onto chromosome 1, resulting in two different der(1)t(1;6) chromosomes observed when DFT2 cell lines are compared. Grey region corresponds to the inactivated centromere from chromosome 6 observed on C-banded chromosomes. Putative breakpoints are indicated by dotted lines and arrows indicated regions undergoing inversion. (l-n) C-banding images of TD_549, TD_500 and TD_554 chromosomes respectively. Arrows indicate non-centromeric staining region on the der(1)t(1;6) chromosome. (**l**,**m**) Scale bar represents 10 μm (same magnification for all images). Abbreviations: DFT2, devil facial tumour 2; pd, population doublings.

**Figure 2 genes-11-00480-f002:**
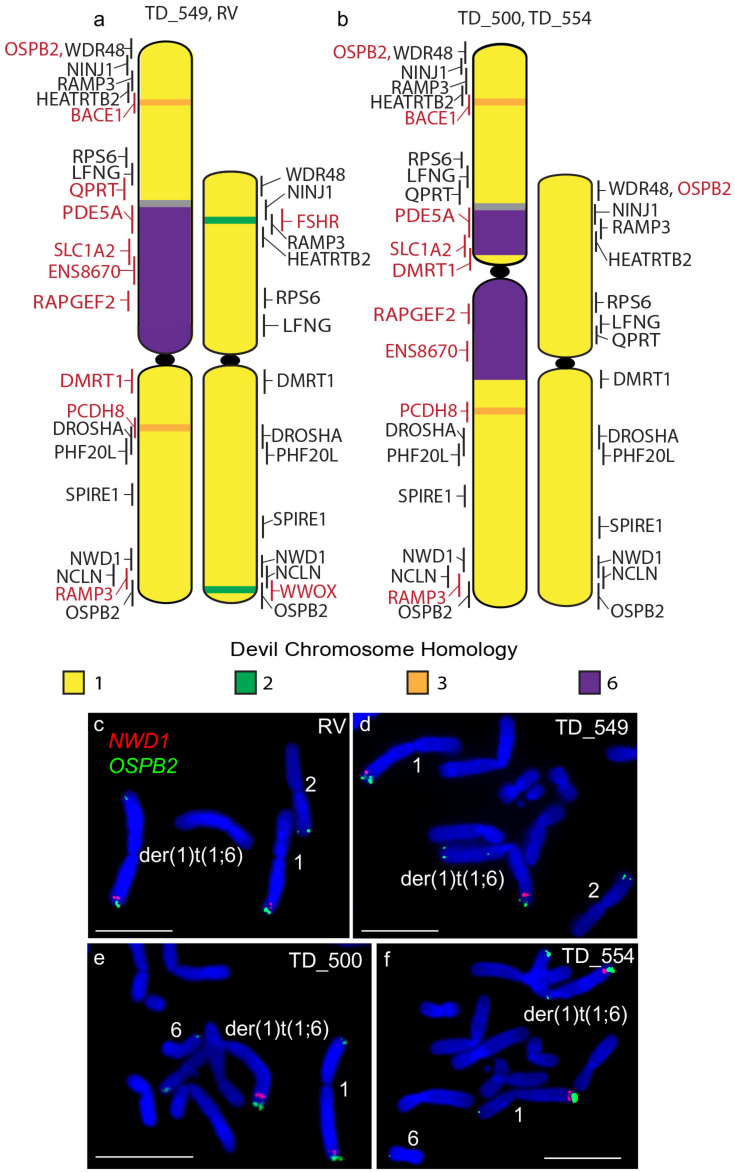
Cytogenetic maps of Chromosome 1 (**a**) TD_549 and RV and (**b**) TD_500 and TD_554. Genes indicated in red differ in copy number between the two copies of chromosome 1 or translocated from a different chromosome. Flpter values (±1 standard deviation) are indicated by vertical lines beside chromosomes. (**c**–**f**) Examples of FISH for genes *NWD1* (red) and *OSBP2* (green) for all four cell lines showing the duplication of *OSPB2*. Scales bar represents 10 μm. Abbreviations: FISH, fluorescence in situ hybridization; Flpter, fractional length p arm terminus.

**Figure 3 genes-11-00480-f003:**
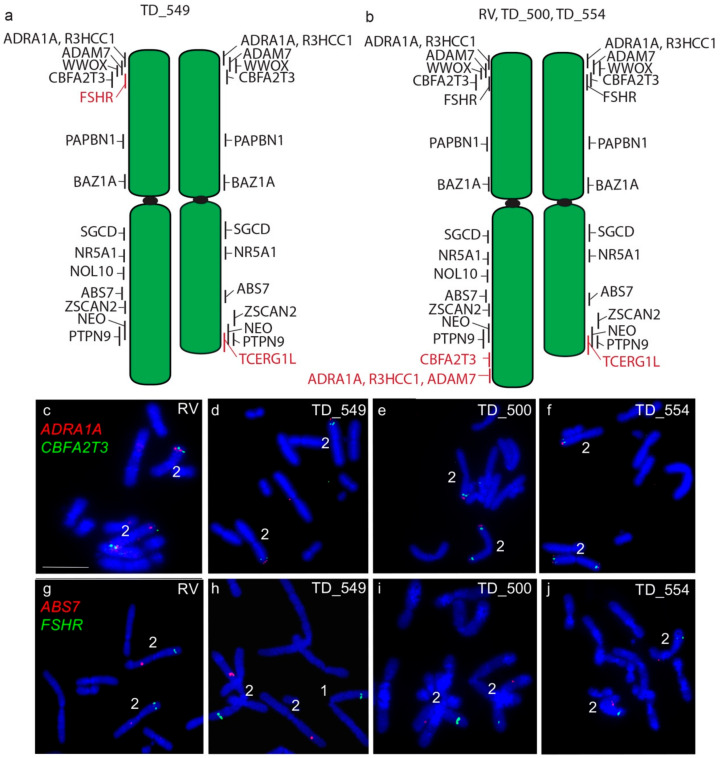
Cytogenetic maps of Chromosome 2 (**a**) TD_549 (**b**) RV, TD_500 and TD_554. Genes indicated in red differ in copy number between the two copies of chromosome 2. Flpter values (±1 standard deviation) are indicated by vertical lines beside chromosomes. (**c**–**f**) In RV, TD_500 and TD_554, *ADRA1A* (red) and *CBFA2T3* (green) are duplicated, mapping to both ends of one copy of chromosome 2, but have not been duplicated in TD_549. (**g**–**j**) In RV, TD_500, and TD_554, *FSHR* (green) and *ABS7* (red) are located on the short and long arms respectively of both chromosome 2 homologues. In TD_549, one copy of *FSHR* has translocated to the short arm of chromosome 1 Scale bar represents 10 μm.

**Figure 4 genes-11-00480-f004:**
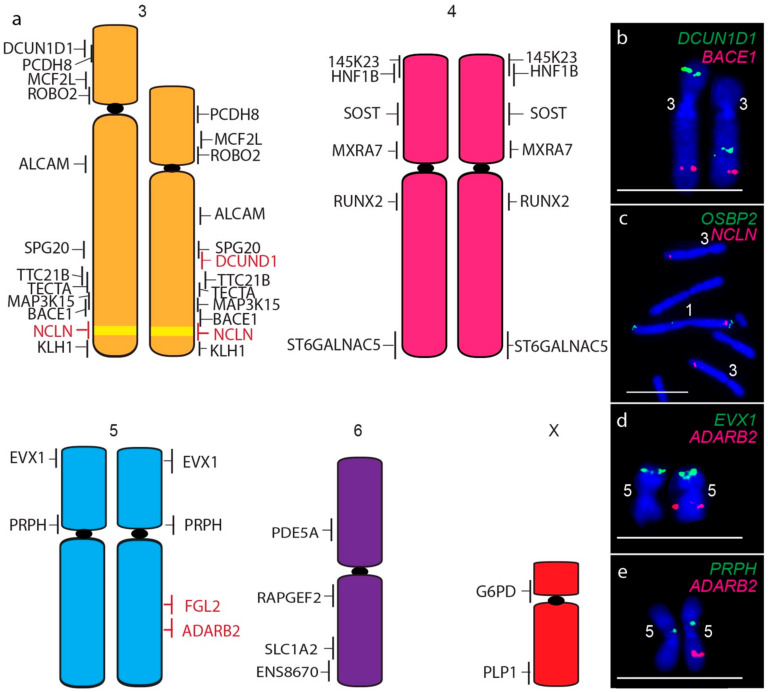
(**a**) Cytogenetic maps of Chromosomes 3 to X. Genes indicated in red differ in copy number between the two copies of chromosome 2. Flpter values (±1 standard deviation) are indicated by vertical lines beside chromosomes. Chromosome 1 material on chromosome 3 is indicated in yellow. (**b**–**e**) Examples of FISH results for genes with differences between chromosome homologues. Scales bar represents 10 μm.

**Figure 5 genes-11-00480-f005:**
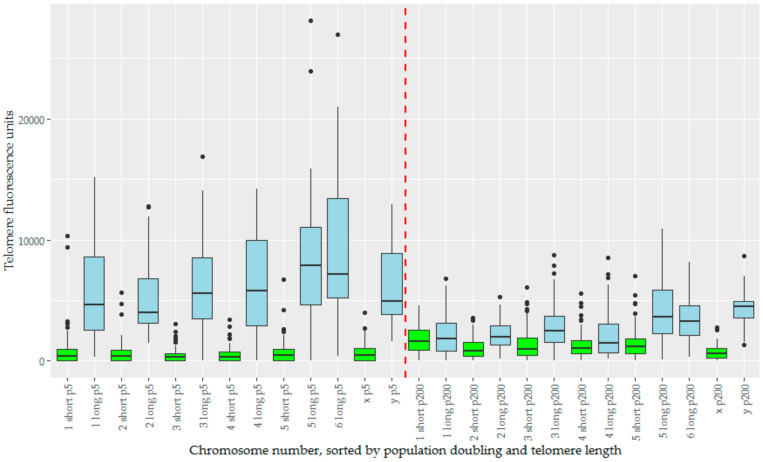
The telomere fluorescence units (TFUs) of individual chromosomes from TD_554 cells sorted by chromosome number, population doubling, and length subset of the telomeres, creating a series of boxplots displaying TFUs for each of these. The long subset of telomeres are represented in blue, and the short subset in green. Population doubling 5 is represented to the left of the red dashed line in the middle, and population doubling 200 to the right of this same line. Within both population doublings, data are sorted in numerical order of chromosomes, ending with the X and Y chromosomes.

**Figure 6 genes-11-00480-f006:**
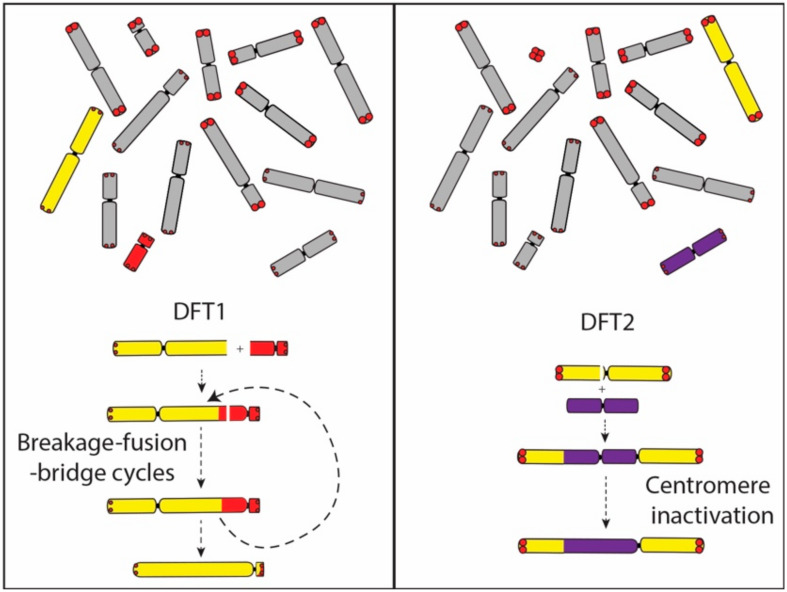
Dicentric chromosome formation in DFT1 and DFT2. In DFT1, the dicentric chromosome formed from the fusion of chromosome 1 and X is proposed to have led to a series of breakage–fusion–bridge cycles until a single active centromere remained. The translocation of chromosome 6 onto chromosome 1 in DFT2 was followed by the inactivation of the chromosome 6 centromere. Abbreviations: DFT1, devil facial tumour 1. DFT2, devil facial tumour 2.

**Table 1 genes-11-00480-t001:** Average change in TFU from pd 5 to pd 200 in each chromosome and length subset, and *p*-values for Games-Howell post-hoc comparisons between pd 5 and 200.

Chromosome	1	2	3	4	5	6	X/Y	Average
Short	789.3	405.5	957.1	743.1	692.6	n/a	22.3 (X)	601.7
*p*-value	0.356	0.799	<0.001	0.003	0.145	n/a	1.000 (X)	-
Long	−3662.7	−2960.4	−3628.9	−4545.4	−4032.5	−5765.1	−1663.7 (Y)	−3787.0
*p*-value	<0.0001	<0.0001	<0.0001	<0.0001	<0.0001	<0.0001	0.131 (Y)	-

**Table 2 genes-11-00480-t002:** Number of telomeres signals below detection limits.

	All Telomeres	Short Subset	Long Subset
pd 5	137 (15.6%)	132 (32.2%)	5 (1.1%)
pd 200	38 (3.9%)	36 (7.9%)	2 (0.4%)

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
