# Peer review of "Comparative Cytogenetic Mapping and Telomere Analysis Provide Evolutionary Predictions for Devil Facial Tumour 2"

_genes, 2020, doi:10.3390/genes11050480_

Round 1

Reviewer 1 Report

The reviewed paper makes important contributions in describing the chromosomal rearrangements that have occurred during the emergence and evolution of the second devil facial tumour disease and tests (in vitro) a specific hypothesis, namely that telomere lengths will become more homogenis over time.

The aims, methods and results were clearly described and the discussion section very adequately details potential limitations and avenues for future work. I have some very minor comments where I believe clarity could be even further improved, but otherwise am happy to recommend it for publication.

Minor Comments:

Line 106: It is not immediately clear that ‘RV’ represents another DFT2 cell line.

Line 119: There is no table 1 presented.

Line 145: specify which three cell lines you mean here.

Line 200: “We mapped genes from this region….” Not clear which region you are referring to here.

Line 202: “was maintained” is repeated in this sentence.

Line 236 (Figure 2 caption): “…between the two copies of chromosome 2…..” Should this be chromosome 1?

Line 237 (Figure 2 caption): Put, “Scale bars represent 10uM” at the end of the caption. At the moment is reads as though it is associated with the upper map.

Line 249 (Figure 3 caption): Put, “Scale bars represent 10uM” at the end of the caption. At the moment is reads as though it is associated with the upper map.

Figure 3 caption: descriptions for panels (e) and (f) are missing

Line 263 (Figure 4 caption): “…between the two copies of chromosome 2…..” Should this be chromosome 1?

Line 277: Define TFU the first time it is used.

Line 462: I think there is a typo in this doi… :)

Section 3.2: Can the differences in TFU between groups be tested in a statistical framework?

Figure 1: What does the orange band in (k) represent?

Figure 2: In the legend for the colours used in the maps, purple should be chromosome 6 I think.

Figure 3: in (b), should ‘TSHR’ on the second homologue read ‘FSHR’?

Figure 4: Indicate with a legend or simply in the caption that the yellow represents chr1 homology. Also, I think you should add the chromosome labels into panels b-e as in the other figures. Including chr1 in panel c.

Reviewer 2 Report

Ingles E and Deakin J provide a descriptive study regarding the chromosome organization of the newly discovered DFTD tumor subtype named DFT2. While the study is very appropriate, informative and helpful in the field in order to understand better the origin and evolution of DFTD, I have several concerns that should be addressed before evaluating its publication

  • In the figure 1 a-f there are only images of TD549 and TD_554 while in the text RV and TD_500 are also mentioned. Authors should provide similar images for these two last DFT2, at least in the supplementary material. Similar situation occurs with Fig 1j that in the text is refer as representative image for TD_500 and TD_554 but the image is only from TD_554. Also, authors are not consisted with the cell lines showing in each experiment. In fig1a-f showed cell lines TD_549 and TD_554 while in Fig 2c-d showed TD_549 and TD_500. Hence, authors should provide images and parallel results from the four cell lines in all the figures since this is a comparative study.

  • I have several concerns when authors state that the broad methylation profile does not change between cell lines and with the number of passages in TD_554. First, all the four cells lines should be shown at pd5 and pd200. Indeed, it is not clear whether TD_500 and TD_549 are in pd5 or pd200. Second, it is not clear which quantification and image analysis the authors used to conclude the similarity in methylation between cells lines. For example, visual observation of the Supplementary figure one shows that the profiles are clearly different between cell lines in chromosome 1 between three cell lines as well as between TD554 pd5 and pd200; in Chromosome 2 between TD_549 and TD_554. This analysis needs a more accurate and quantitative analysis. The chromosomes X and Y are difficult to see in the image. I recommend enlarging the imaging of these particular chromosomes. I would also suggest showing analysis of DFT1 and devil fibroblasts although the authors provide some references.

  • For the comparison of the progression of the length of telomeres over the number of passages, authors remain not showing the results on the four cell lines object of the study. Indeed, in the legend of the figure 5 did not indicate which cell line are the results from. Authors should show the progression of the telomere length in four cell lines over 200pd and compare to DFT1-5pd/200pd to see whether DFT2 telomeres short to the same level.

  • The homogenization of the telomeres might occur do to the process of culturing the cells in a 2D models which will argue against a possible mechanism of evolution in DFTD DNA organization. I believe that for sustaining their postulations authors should examine telomere length of DFT2 at 5pd and 200pd in comparison to DFT1 in a matrigel system, or forming spheroids or in models of tumor xenografts

Minor changes:

  • Describe in the text the list of genes mapped from the region of chromosome in long culture TD_554.
  • Fix the line of text 202 and 208

Round 2

Reviewer 2 Report

The authors have satisfactory answered my comments.